# Impact of Improper Storage of ChAdOx1-S (AstraZeneca) Vaccine on Its Efficacy and Safety

**DOI:** 10.3390/vaccines11010093

**Published:** 2022-12-30

**Authors:** Marek Mikołajczyk, Roman A. Lewandowski, Anatoliy G. Goncharuk

**Affiliations:** 1Allergology Department of the Voivodeship Rehabilitation Hospital for Children in Ameryka, 11-015 Olsztynek, Poland; 2Institute of Management and Quality Science, Faculty of Economics, University of Warmia and Mazury in Olsztyn, 10-720 Olsztyn, Poland; 3Hauge School of Management, NLA University College, 4633 Kristiansand, Norway

**Keywords:** ChAdOx1-S (AstraZeneca) vaccine, COVID-19 vaccination, cold supply chain, vaccine efficacy, vaccine safety

## Abstract

Background: In May 2021, there was an incident regarding giving patients AstraZeneca vaccines stored improperly. They were stored at room temperature (21 degrees centigrade) for 18 h, 12 h longer than the producer recommends. Aim of the study: The paper aims to contribute to the body of knowledge concerning the efficacy and safety of the ChAdOx1-S (AstraZeneca) vaccine concerning the requirements for cold supply chain specification. Patients and methods: Improperly stored vaccines were given to 44 patients, and 39 of them decided to take part in the study. The Control group consisted of 56 people vaccinated on the same days by the same medical teams, using properly stored medicines. Results: The concentration of anti-S1 SARS-CoV-2 Spike protein IgG antibodies did not differ significantly between the groups. Examined group median 70 kU/L (20;100). Control group median 66 kU/L (32.75;100), *p* = 0.751. We did not observe any COVID-19 infections in either the control or examined group for half a year after the incident. People from each group reported that local and systemic adverse events occurred directly after the first and second doses. In the control group, one case of spontaneously subsiding face edema and joint pain was observed. There were no severe or fatal adverse events. There were no significant differences between the groups, besides the fatigue, after the second dose. Conclusion: AstraZeneca vaccine ChAdOx1-S stored at 21 degrees centigrade for 18 h before vaccination has the same safety profile (*p* < 0.05) and the same efficacy (*p* < 0.05) as the vaccines stored in conditions recommended by the producer.

## 1. Introduction

At the end of 2019, the COVID-19 pandemic broke out first in China and gradually spread all over the world. The plague spread quickly, leading to a large number of hospitalizations and deaths. There were no known ways of treating this new disease, and many countries lapsed into prolonged lockdowns. Humanity was then confronted for a long time with an unseen medical hazard. The scientific community started a frantic race to recognize the infection factor named SARS-CoV-2 and to find an effective treatment. Scientists acted on vaccine development. There were many attempts to create a working preparation. The first working products were based on the mRNA coding virus Spike protein closed in a lipid capsule. The Moderna and Biontech-Pfizer vaccine brought promising results in clinical trials and were introduced for common COVID-19 prevention in the developed world [1,2]. Scientists trying to use harmless viruses as vectors to transmit information about virus proteins into human cells also noted successes. Oxford University and AstraZeneca used the chimpanzee airway virus to activate the human immune system against the SARS-CoV-2 Spike protein with good effect [3]. Johnson&Johnson produced a well-functioning vaccine based on the same idea and used modified human adenovirus 26 [4]. There were other projects based on concepts used earlier in vaccines produced against other diseases. Experiments with incapacitated SARS-CoV-2 virus led to a vaccine created by Sinopharm (China) [5]. Novavax developed a vaccine based on nanoparticles [6]. These products were quickly introduced for mass vaccination.

Since the vaccines against COVID-19 were developed, the vaccination action started in many countries. The pharmaceutical companies increased production, and mass vaccination points were created. The way of delivering the medicines was complicated not only due to special conditions that had to be fulfilled but also to the variety of these conditions for different vaccines. Deep freezing and different acceptable periods of storing at room temperature for each vaccine before giving it to patients additionally complicated the process and sometimes led to mistakes.

The success of a vaccination program is based not only on the percentage of vaccine effectiveness but also on cold supply chain management since failure in this field may lead to a situation that the high effectiveness becomes wasted because of the temperature-sensitive nature of vaccines [7,8,9]. Even though vaccines are stored at cold temperatures to maintain their stability and immunogenicity, they lose their potency over time, but higher temperatures accelerate this potency loss process [10]. The improper storage can influence vaccine stability, for example, measured by the fluorescence focus assay of viral titers [11], leading to reduced efficacy and changing the safety profile [12,13,14]. It is important to note that low temperatures can also result in potency losses, e.g., inactivation of aluminum adjuvanted vaccines due to freezing [15].

To limit the risk of vaccine safety and efficacy being compromised during transport and storage, very restrictive requirements may be introduced in this regard. However, on the other hand, applying excessive conditions for cold supply chain management may result in high costs and the need to dispose of vaccines when established storage conditions are broken. The determination of optimal conditions ensuring the safety and efficacy of a vaccine in relation to the storage and cold chain supply requirements is very difficult due to vaccines’ macromolecular complexity. “It has not been possible to date to identify a physicochemical assay that can be directly related to the vaccine’s potency. Thus, biological assays (i.e., in vivo animal immunogenicity tests or in vitro cell-based or antibody binding-based assays) are the cornerstone of vaccine potency and stability testing” (p. 244) [15]. AstraZeneca (ChAdOx1-S) vaccine uses the same platform as the ChAdOx1-GnGc Rift Valley Fever vaccine, and the recommendations for the ChAdOx1-S vaccine cold chain were derived from the predecessor [16,17]. However, vaccine heat sensitivity and stability profiles vary depending on the manufacturer regardless of the same platform [16]. However, we did not find any studies concerning directly ChAdOx1-S potency loss due to the vaccine cold chain. Therefore, conducting research related to the search for optimal storage conditions for this vaccine is very important. Additional sources of knowledge concerning the safety and efficacy of the vaccine could be also the collection of all incidents, allowing to check a vaccine’s impact on adverse events and immunity.

The paper aims to contribute to the body of knowledge concerning the efficacy and safety of the ChAdOx1-S (AstraZeneca) vaccine in relation to the requirements for cold supply chain specification. In particular, the study supports to some extent the overall goal of a vaccine stability program during clinical development, such as to confirm that a vaccine remains within the boundaries of the upper and lower potency limits throughout its shelf-life (Krause, 2009). This study is expected to facilitate technology developers, logistics managers, and policymakers in the process of vaccination program development and implementation.

In the early spring of 2021, Polish patients received the possibility of vaccination using the AstraZeneca vaccine. There were many vaccination points created. The producer recommends storing the vaccine at 2–8 degrees centigrade and warm to a temperature below 30 degrees for no longer than 6 h before vaccination. The situation of mass vaccination, performed often by not well-prepared facilities and well-trained staff, was a real challenge in many places. In May 2021, in a Polish hospital, there was an incident regarding giving patients improperly stored AstraZeneca vaccines. They were stored at room temperature (21 degrees centigrade) for 18 h, a week before vaccination. When the problem was discovered, the accident was reported to the appropriate authority, and the decision was taken to check how the event influenced the vaccinated patients’ health. The knowledge about the stability of vaccines, determined by the producer, was that they could be stored for no longer than 12 h in temperatures no higher than 27 degrees centigrade. Therefore it was also anxiety about the possible adverse events in the treated population.

The frequency of the side effects connected with each vaccine usage was established, and besides mild problems such as site-vaccination pain, chill, fatigue, fever, and diarrhea, severe adverse events were seen as very rare. The rate of myocarditis was 4.8 cases per 10000 vaccinated adolescents using mRNA products in the Israeli population. All cases were mild and hemodynamically stable, and only some of them needed short hospitalization [18]. Other, such as anaphylactic shock, was also described, but the frequency of the event, in some trials, did not differ from the placebo group [19]. Shultz N. described thrombosis and severe thrombocytopenia in five healthcare workers after ChAdOx1-S vaccination. Four of the patients had major cerebral hemorrhages [20]. Greinacher A. described 11 German and Austrian patients with the same adverse event after the same vaccine [21]. Pottegard A. analyzed events of thrombosis in Denmark and Norway ChAdOx1-S vaccinated population and observed 59 venous thromboembolic events compared with 30 expected based on the incidence rates in the general population [22]. Hippisley-Cox J. described an increased risk of thrombocytopenia and thromboembolism after ChAdOx1-S and BioNTech mRNA vaccination [23]. Introna A. described a 62-year-old caucasian man with Guillain-Barré syndrome after AstraZeneca COVID-19 vaccination [24]. Young Gi Min described similar cases in the Korean population [25].

## 2. Patients and Methods

Improperly stored vaccines were given to 44 patients. For 25 people, it was the first dose and for 14, the second dose in the 2-dose vaccination scheme recommended at that time by the Polish Mistry of Health. The recommendation for ChAdOx1-S (AstraZeneca) was to give a patient 2 doses 12 weeks apart. Patients who were given an improperly-stored vaccine as the first dose were vaccinated the second time using properly stored vaccines after 12 weeks. All of the patients who were given one improperly-stored vaccine (first or second dose) were invited to the research study, and 39 out of 44 decided to take part in the study. Thus, during the research, all participants were “fully” vaccinated, according to the national guidelines. To build a control group, invitations to 93 people vaccinated on the same days by the same medical teams using properly stored medicines were issued. In total, 56 of them positively answered the invitation and participated in the study. Sampling was conducted in such a way that the time from the second dose of vaccine to the time of the study was identical for all participants in both groups. This way of sampling, in the authors’ opinion, was the most appropriate to build an adequate control group, which would be, to the greatest extent possible resistant to disturbing factors that could make the group not random (Table 1).

### 2.1. Questionnaires

All patients were examined and interviewed by an experienced physician six weeks after the second dose. They filled out a questionnaire about adverse effects, chronic diseases, and past COVID-19 confirmation.

### 2.2. Measurement of Antibodies against SARS-CoV-2 Phosphorylated Nucleocapsid Protein

Antibodies of IgG class were measured in serums six weeks after the second dose, using Polycheck tests (Germany). The measured range was 0.15–100 kU/L, and results higher than 0.7 kU/L were considered positive. The measurements were performed to ensure that the patients’ anti-SARS-CoV-2 immune status did not differ between the groups before vaccination.

### 2.3. Measurement of Antibodies against SARS-CoV-2 Subunit S1 Spike Protein

The immune answer to the vaccination was evaluated by checking the patients’ anti-S1 IgG serum levels. Antibodies were measured in serums six weeks after the second dose, using Polycheck tests (Germany). The measured range was 0.15–100 kU/L, and results higher than 0.7 kU/L were considered positive.

### 2.4. Statistics

Statistical analyses were performed using PQStat software. Mann-Whitney U tests were used to compare antibody concentrations between the groups. Demographic data were compared using the t-Student test for age and chi-square tests for other parameters. Adverse events were compared using chi-square tests.

## 3. Results

### 3.1. Characteristics of the Samples

Table 1 shows the characteristics of 39 participants of the examined group and 56 of the control group. In both groups, there were more males (55% and 59%). However, the differences are not statistically significant (*p* = 0.726 for Pearson Chi-square). Similarly, the difference between chronic diseases occurring in the studied groups, their age, and passed COVID-19 are not statistically significant. The conducted interviews and surveys in terms of chronic diseases show that in the examined group, three persons had hypertension only, two hypertension and diabetes mellitus, two ischemic heart disease, one hypertension and Hashimoto disease, one diabetes mellitus only, one arrhythmia only, one multiple sclerosis only, and one COPD only. The control group patients had four persons with hypertension only, three with hypertension and diabetes mellitus, four with Hashimoto disease, one with podagra, two with asthma, one with arrhythmia, one with venous insufficiency, one with rheumatoid arthritis, one with cerebral palsy, one with normotensive hydrocephalus, one with allergic rhinitis, one with diabetes mellitus and arrhythmia, and one with ischemic heart disease.

### 3.2. Efficacy

The concentration of anti-S1 SARS-CoV-2 Spike protein IgG antibodies did not differ significantly between the groups. Examined group median 70 kU/L (20;100), mean 60.29 ± 39.6. Control group median 66 kU/L (32.75; 100), mean 62.82 ± 35.4, *p* = 0.751. (Figure 1). We have not observed any COVID-19 infections in either the control or examined group for half a year after the incident.

### 3.3. Adverse Events

People from each group reported that local and systemic adverse events occurred after the first and second doses (Figure 2). After the first dose of the vaccine, fever was reported in 41% of the examined and 50% of the control group, headache in 39.5% and 41.1%, respectively, nausea/diarrhea at 2.6% and 3.6%, fatigue at 38.5% and 53.6%, and injection site pain in 61.5% and 62.5%. The second dose of the vaccine was connected with fever in 2.6% of the examined group and 12.5% of the control group, headache in 5.1% and 16.1%, respectively, nausea/diarrhea in 0.0% in both groups, fatigue in 7.7% and 23.2%, injection site pain in 28.2% and 37.5%. In the control group, one case of spontaneously subsiding face edema and joint pain was observed. There were no severe or fatal adverse events.

## 4. Discussion

The groups of patients included in the study did not differ in demographic characteristics. The COVID-19 status of examined persons was analyzed to minimize the influence of previous contact with the SARS-CoV-2 virus on the efficacy and adverse events after the vaccination process. In addition to the interviews, the concentration of antibodies against SARS-CoV-2 phosphorylated nucleocapsid protein, which is not generated by AstraZeneca vaccination was measured. We did not observe significant differences between the groups.

The improper storage can influence vaccine stability leading to reduced efficacy and changing the safety profile [12,13,14]. There was concern about serious complications in our patients, but they, fortunately, did not happen. Patients reported mild and moderate adverse events in both groups and after each dose of the vaccine. The analysis of the first dose showed a similar frequency to all collected parameters. The second dose of the vaccine showed small differences in adverse events between the groups. Patients from the group vaccinated using badly stored vaccines better tolerated the vaccination. The study showed that the frequency of local reaction was similar, but the percentage of people who felt fatigued after the second dose was a little higher in the control group (*p* = 0.048). The frequency of fever, headache, and diarrhea was similar. The comparison of the frequency of the adverse events observed in the study and described in other clinical trials is similar. The phase three study of efficacy and safety of ChAdOx1-S vaccine showed adverse events rate: headache-50.2%, fatigue-49.7, nausea-15.3%, fever-7%, injection site pain-58.3% [26]. The frequency of a fever reported by our study participants was higher than described above. We cannot explain this difference. Maybe this is the individual reaction of this small group of people taking part in the study. The frequency of the fever in both the examined and control groups was statistically indifferent, so it did not influence the study results and conclusions.

The significant difference, in our study, between the examined and the control group in fatigue, after the second dose is very difficult to explain according to this study. It might be a result of less stable virus factors of the vaccine, but it seems impossible, in our opinion, to take into account other presented factors. The frequency of other adverse events was insignificantly different between the groups. The efficacy was understood as the median of anti-S1 antibodies concentration, and the frequency of morbidity was also similar. Change in only one parameter seems to be too small a premise to conclude that.

The time of IgG anti-S1 Spike protein measurement in the sixth week after vaccination was chosen as the best, based on previous clinical trials [27,28]. The immune answer to vaccination using improperly stored vaccine was similar to the control group. The median of anti-SARS-CoV2 S1 antibodies serum concentrations did not differ significantly. There were no SARS-CoV-2 infections observed in either of the groups during the study period.

The limitation of the study is the small examined group size. The number of patients was a result of the course of the accident and was regardless of our decision. The stability of the badly stored vaccines was not checked using laboratory tests because we did not have at our disposal the tools to perform the appropriate investigation.

## 5. Conclusions

The determination of the recommendations about the storage and cold chain supply conditions that ensure a vaccine remains within the boundaries of the upper and lower potency limits throughout its shelf-life is very difficult due to the vaccines’ macromolecular complexity [15]. This largely excludes the possibility of identification of the vaccine’s potency directly through a physicochemical assay. Biological assays (i.e., in vivo animal immunogenicity tests or in vitro cell-based or antibody binding-based assays) are the foundation of vaccine potency testing since testing on humans creates many more challenges. Therefore, our results based on the study of a group of humans might be an important contribution to the body of knowledge concerning testing the efficacy and safety of the vaccine in relation to the vaccine cold supply chain. The research shows that the AstraZeneca vaccine ChAdOx1-S stored at 21 degrees centigrade for 18 h before vaccination has a similar safety profile (*p* < 0.05) and comparable efficacy (*p* < 0.05) as the vaccines stored in conditions recommended by the producer. This leads us to postulate that the storage requirements for the vaccine could be adjusted. Less strict requirements could help in the distribution and storage in countries where the cold chain is difficult and sometimes impossible to achieve and also reduce vaccine losses resulting from improper (according to the strict recommendations) transport and storage.

## Figures and Tables

**Figure 1 vaccines-11-00093-f001:**
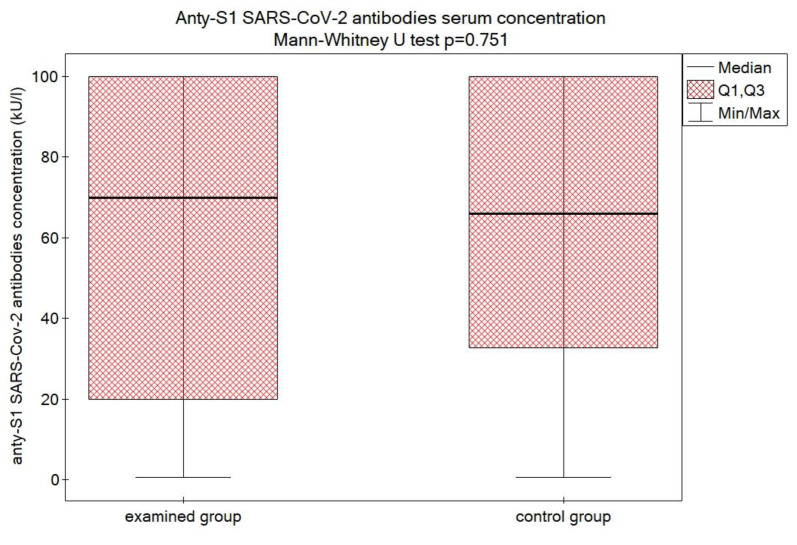
The comparison of the concentration of anti-S1 SARS-CoV-2 Spike protein IgG antibodies in the examined and control group.

**Figure 2 vaccines-11-00093-f002:**
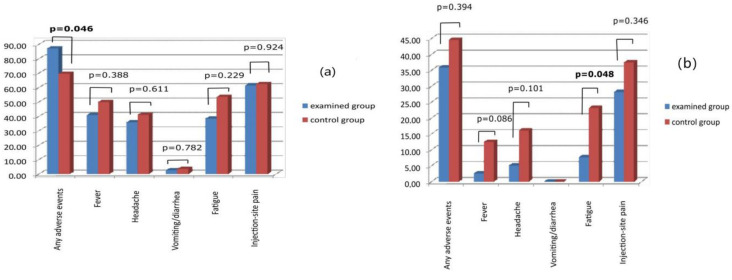
Adverse events after the: (**a**) first dose of vaccine ChAdOx1-S, (**b**) second dose of vaccine ChAdOx1-S.

**Table 1 vaccines-11-00093-t001:** Characteristic of samples.

Variables		Examined Group	Control Group	*p*-Value
**Sample Size**		39	56	
Gender	Male (%)	23 (55%)	31 (59%)	0.726
	Female (%)	16 (45%)	25 (41%)	
Age (years, mean ± SD)		44.38 ± 18.10	45.77 ± 17.32	0.706
Chronic diseases		12	22	0.394
Confirmed COVID-19 in the past		10	12	0.632
Anti-N antibodies concentration (kU/L) (median (25; 75 pc))		0.54 (0.3; 1.65)	0.33 (0.27; 0.91)	0.078

## Data Availability

Data from the study are available at: http://dx.doi.org/10.13140/RG.2.2.29048.78080 (accessed on 2 December 2022).

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
