# Peer review of "Impact of Improper Storage of ChAdOx1-S (AstraZeneca) Vaccine on Its Efficacy and Safety"

_vaccines, 2022, doi:10.3390/vaccines11010093_

Round 1

Reviewer 1 Report

I did not observe any problems in the manuscript and the subject is extremely relevant.

Author Response

No comments.

Reviewer 2 Report

Your paper Does not fit minimum quality of scientific evidence on vaccine stability.

The summary does not present the results and conclusion!!

There is no background review to show  knowledge gaps and needs for research. What would be the contribution of such a study to improve  the vaccine storage criteria?

The discusssions do not deal with the methodology limits and do not lead to improvements of vaccine producer recommendations.

There is no conclusion report.

However, this study witnesses a legitimate concern of the hospital managers

about the efficacy of the  vaccine used in not optimal storage conditions. 

Author Response

Reviewer #2

The summary does not present the results and conclusion!!

Response: Agree. We have thoroughly rewritten the manuscript in order the results and conclusions are relevant to the design of the research.

There is no background review to show knowledge gaps and needs for research. What would be the contribution of such a study to improve the vaccine storage criteria?

Response: Agree. We extend the background review to show the gaps and needs for research and describe the possibility to improve the vaccine storage criteria.

The discussions do not deal with the methodology limits and do not lead to improvements of vaccine producer recommendations.

Response: Agree. We described the limits of the methodology and propose improvements of vaccine producer recommendations.

There is no conclusion report.

Response: Agree. We extended the conclusion section in the manuscript.

However, the study witness a legitimate concern of the hospital managers about the efficacy of the vaccine used in not optimal storage conditions.

Response: Agree. The adverse event showed problems with vaccine storage procedures in the hospital and to our knowledge, these procedures have been improved and staff has been retrained at this hospital. But on the other hand, this adverse event allowed us to perform the research presented in the manuscript and propose to the manufacturer some recommendations concerning the improvement of the storage procedures of the vaccine.

Reviewer 3 Report

In this case report entitled "Impact of improper storage of ChAdOx1-S (AstraZeneca) vaccine on its efficacy and safety", authors explained and analyzed the effect of improperly stored AstraZeneca vaccines given to the 44 people. 

there are few concerns as follows:

1. The abstract should be revised to include the results/impact of the immunization as well as the overall conclusion of this study.

2. Line no. 30-31 "For 25 people it was the first dose, for 14 the second dose. All of them were fully vaccinated and got one dose of properly-stored vaccine."  this is not clear to me. If for 25 people it was first dose, the how they got one dose of properly-stored vaccine. please explain.

3. Tables and figures should be adjusted and formatted according to the journal's guidelines.

Author Response

Reviewer #3

The abstract should be revised to include the results/impact of the immunization as well as the overall conditions of the study.

Response: Agree. The abstract was revised to include the results/impact of the immunization and the overall conditions of the study.

Line no. 30-31 “For 25 people it was the first dose, for 14 the second dose. All of them were fully vaccinated and got one dose of properly-stores vaccine.” This is not clear to me. If for 25 people it was the first dose, how they got one dose of properly-stored vaccine? Please explain.

Response: Agree. The clarity of the line was improved as follows: “For 25 people it was the first dose and for 14 the second dose in the 2-dose vaccination scheme recommended at that time by the Polish Mistry of Health. The recommendation for ChAdOx1-S (AstraZeneca) was to give a patient 2 doses 12 weeks apart. Patients who were given an improperly-stored vaccine as the first dose, in 12 weeks were second time vaccinated by the properly stored vaccine. All of the patients who were given one improperly-stored vaccine (first or second dose) were invited to the research and 39 out of 44 decided to take part in the study. Thus, during the research, all participants were “fully” vaccinated, according to the national guidelines.”

Tables and figures should be adjusted and formatted according to the journal’s guidelines.

Response: Agree. Tables and figures were adjusted and formatted according to the journal’s guidelines.

Reviewer 4 Report

Summary:

The manuscript titled “Impact of improper storage of ChAdOx1-S (AstraZeneca) vaccine on its efficacy and safety” evaluates the efficacy and safety concerns related to the use of improperly stored COVID vaccine given to a subset of the population in Poland. Overall, the study explores a very important topic regarding the use of approved COVID vaccines and maintaining required cold chains for vaccine storage.  But more details on experimental design and a deeper discussion is needed in the manuscript draft. Additional data is needed to justify that improper storage of the vaccine does not affect immunogenicity and efficacy of the vaccine short term and long term. Please review the text for consistent formatting and grammar errors.  

I have edits/comments to share:

Patients and Methods

How were the 56 control patients enrolled? Was it strictly by invite only? Did these patients get screened at all?

Results

The first chart has a chronic diseases category. Can the authors define this category? Why was this category included in the study?

Discussion

The authors note that the second dose of vaccine in the poorly stored vaccine group had a better tolerance of the vaccine. Can the authors comment on why they believe this occurred? 

While it is recommended to have a cold chain for this vaccine, there is no discussion or analysis on what happens to the vaccine if stored improperly. Do the vectors within the vaccine get destroyed and become useless at higher temperatures? What occurs with the composition of the vaccine when stored at warmer temps? Such information can help explain the results of the study. 

Author Response

Reviewer #4

The manuscript titled “Impact of improper storage of ChAdOx1-S (AstraZeneca) vaccine on its efficacy and safety” evaluates the efficacy and safety concerns related to the use of improperly stored COVID vaccine given to a subset of the population in Poland. Overall, the study explores a very important topic regarding the use of approved COVID vaccines and maintaining required cold chains for vaccine storage.  But more details on experimental design and a deeper discussion are needed in the manuscript draft.

Response: Agree. More discussion about the experiment design was added and overall the manuscript was extended and improved.

Additional data is needed to justify that improper storage of the vaccine does not affect the immunogenicity and efficacy of the vaccine short term and long term.

Response: Agree. Additional data was added.

Please review the text for consistent formatting and grammar errors.  

Response: Agree. The text was proofread.

Patients and Methods

How were the 56 control patients enrolled? Was it strictly by invitation only? Did these patients get screened at all?

Response: More details about the study design was added to the manuscript.

Results

The first chart has a chronic disease category. Can the authors define this category? Why was this category included in the study?

Response: In the study were included all chronic diseases reported by the participants. The chronic diseases are discribed in the manuscript.

Discussion

The authors note that the second dose of vaccine in the poorly stored vaccine group had a better tolerance of the vaccine. Can the authors comment on why they believe this occurred? 

Response: Agree. We comment on that issue in the discussion section.

While it is recommended to have a cold chain for this vaccine, there is no discussion or analysis on what happens to the vaccine if stored improperly. Do the vectors within the vaccine get destroyed and become useless at higher temperatures? What occurs with the composition of the vaccine when stored at warmer temps? Such information can help explain the results of the study.

Response: Agree. We add information on what happens to the vaccine if stored improperly.

Round 2

Reviewer 2 Report

I appreciate your improvements.

Given the storage constraints in many countries and regions over the world, the results of this study are useful to know. So, i suoort tyour publication

Of course, more studies on the efficacy impact of various storage conditions are needed.

Reviewer 4 Report

The authors did a nice job addressing the recommended edits.